# The Impact of Selected Atmospheric Conditions on the Process of Abrasive Wear of CFRP

**DOI:** 10.3390/ma13183965

**Published:** 2020-09-08

**Authors:** Aneta Krzyzak, Damian Racinowski, Robert Szczepaniak, Mateusz Mucha, Ewelina Kosicka

**Affiliations:** 1Faculty of Aeronautics, Military University of Aviation, ul. Dywizjonu 303 nr. 25, 08-521 Deblin, Poland; r.szczepaniak@law.mil.pl (R.S.); m.mucha@law.mil.pl (M.M.); 233rd Airlift Base, ul. Witkowska 8, 62-430 Powidz, Poland; racinowicz@gmail.com; 3Faculty of Mechanical Engineering, Lublin University of Technology, ul. Nadbystrzycka 36, 20-618 Lublin, Poland; e.kosicka@pollub.pl

**Keywords:** tribology, CFRP composites, weathering, thermal shocks

## Abstract

The aim of this study was to examine the impact of weathering and thermal shocks on the abrasive wear of epoxy resin composites reinforced with carbon fabric that are commonly used in aviation. The composite was exposed to degradation in an apparatus simulating weathering and thermal shocks and then subjected to an abrasion process, with and without the presence of water. The abrasive wear was controlled by checking the weight loss as well as by visual inspection. The research findings indicated a significant effect of the presence of water in the process of friction upon the deterioration of composite resistance to abrasion with regard to dry friction. The long-term impact of rapid cyclic temperature changes (temperature difference: from −56.5 °C to +60 °C) and a combined effect of UV-A (0.83 W/m^2^), along with condensation of vapor and an increased ambient temperature (above 50 °C), influenced an improvement in resistance to abrasive wear. The environment of thermal shocks diminished abrasive wear to a much smaller extent than after exploitation in an environment of weathering but both environments contributed to the degradation of the surface layer. Additionally, the environment with UV-A radiation resulted in exposure of the composite reinforcement already after four months of environmental impact.

## 1. Introduction

Over the last half century, the aviation industry has evolved at an extremely fast pace. The exploited technologies and materials are of the highest quality. Moreover, manufacturers have focused on the light weight and excessive strength of plastics used in aviation [1]. The answer to the need of achieving the lightest airplane that retains the assumed strength while meeting all safety standards is polymer composites, which are usually used as a replacement for heavier metals (for example, a duralumin skin may be replaced by a composite reinforced with carbon fibers [2,3].

Increasing the share of epoxy resins and fabric reinforcements in aircraft construction contributes to conducting numerous and detailed studies of newly designed materials. Weight reduction, despite obvious economic benefits, can have a negative effect on the strength and the lifespan of aircraft components exposed to hostile working conditions. The conducted investigations must therefore include environmental conditions in which the components made of composite materials will be ultimately exploited. An example is an aggressive environment that accelerates thermal degradation or where the heat accelerates fatigue degradation [4,5,6,7,8].

Reflecting on polymer composite damage, it should be noted that the processes of matrix base destruction are linked with the phenomena occurring in-between each phase. Most often, it is the loss of adhesion between the reinforcement and the polymer. Changes occurring in composites made up of polymer materials is exerted by chemical and physical changes, occurring in the processing, storage, and exploitation [9].

Due to an increasingly wider use of composite materials for the production of machine components [8], it should be noted that the performance of machinery and devices correlates with maintaining the reliability of the weakest link, which usually turns out to be a sliding pair or a rolling pair [6].

Therefore, it is necessary to determine the effect of the composition of the materials [10,11] as well as the working conditions (such as temperature, UV radiation and humidity) on the properties for parts made of these materials.

In the conducted literature analysis, many examples of studies of tribological properties of composite materials were found. The examples described below and grouped in Table 1 relate to issues related to the composition of composites, their production method, and the influence of external factors.

The authors conducted research in which they determined the impact of various physical modifiers on the tribological properties of polymer composites. These modifiers include modifiers such as Al_2_O_3_, TiO_2_, ZnO, CuO, SiC, ZrO_2_, Si_3_N_4_, SiO_2_, or CaCO_3_ [12,13,14,15]. The frictional coefficient of the composites with friction modifiers was compared with the neat matrix up to a filler fraction (e.g., [16,17,18,19,20]). In addition, the obtained results indicated that particles could obviously reduce the value of the frictional coefficient, even at low filler loadings. The influence of particulate fillers on the stress-strain behavior of polymers can be determined when the composition changes are applied.

Tribological studies have also been conducted in terms of the effect of allotropic forms of the additives used. The element added to composites in various allotropic forms is carbon. The publications present the results of tribological studies in which graphite [16,17,18], carbon nanotubes [16,19], graphene [18,19,20,21] or fullerene [22] have been used as additions to composites. Promising research from the results show great interest in these supplements.

Shaping the tribological properties with particles of various sizes is also well known [23,24,25,26]. The authors repeatedly point out that reducing the particle size to nanoscale (<100 nm) causes a change in the consumption efficiency of these nanocomposites from the performance of systems filled with micron particles [27]. It should be noted that such modifiers cause the presence of the additive in agglomerations, which is an eliminated phenomenon. Aspects related to the composition of composites concern not only the modifiers themselves, but also the reinforcement used in the form of fabrics [28,29,30,31,32].

In addition to modifying the composition of the composite, researchers are trying to link the method of their production with the obtained properties (including tribological ones) [33,34]. Aspects of changes in composite production technology are therefore another research path that should be developed. This will allow further definition of mechanisms affecting the properties of final products.

However, other factors affect the abrasive wear of polymer composites. It is also important to determine how environmental factors change their tribological properties. This matters when we analyze the reliability of the responsible parts of machines and devices made of composites that work in demanding conditions. These are issues related to, among others with loads in which composite parts work, but also factors such as humidity or temperature [12,35].

The above-mentioned examples show the nature of research conducted so far and are designed to indicate the trend of research news with regard to the conducted examinations over the abrasive wear of composite materials subjected to various modifications and operating conditions. The growing interest in composite materials, which is a result of continuous improvement in their composition and properties requires additional analysis of accelerated tests on the influence of the environment on different properties. This translates into the appearance of publications considering, amongst others, aspects of tribology. The analysis of the available literature regarding the abrasion of composite materials has allowed for focus on the possibilities of supplementing previously unexamined areas. Despite a lot of research in the tribology of polymer composites, there are no results in the literature that show the influence of thermal shocks as an environment on the tribology properties.

The assessment of abrasive wear is possible thanks to the analysis of the topography of sliding surfaces. Profilometry provides a quantitative measure of the surface roughness. In profilometry, a probe, mechanical or optical, is passed across the surface and follows the contours at each point on the surface, and the height of the probe at each point is recorded and the resulting 1D scan or a 2D map is analyzed. Parameters such as arithmetic average of the absolute values of all points of the profile (Ra) and root means square values of all the heights around the mean (Rq) are often used to determination the roughness. These, however, do not reflect the distance between the features and their shape [36]. The literature on the sliding surface analysis of polymer composites suggests investigating the topography of worn surfaces by scanning electron microscopy (SEM) [37] or by optical microscopy [38].

The aim of this paper was to examine the impact of selected atmospheric conditions occurring on the process of abrasive wear of carbon fiber fabric-reinforced epoxy composites, which are most often used in aviation. In real operating conditions of plastics, UV radiation, humidity, temperature, and chemical substances interact with one another and accelerate the destruction processes. Superficial characterization methods were used to understand abrasive wear.

## 2. Research Methods

### 2.1. Materials

The research into abrasive wear was carried out on composites used in aviation. In the manufacture of the composite, the authors used epoxy resin L 285 together with dedicated hardeners H 286 and H 287 as well as a carbon fiber fabric GG 280 P/T. Both hardeners, in accordance with the manufacturer’s recommendations, were in equal proportions. The ratio of the resin to the hardeners equaled 100:40. Epoxy resin L 285 is intended for laminating at room or higher temperatures and was designed to produce composite structures that are subjected to significant loads (e.g., aircraft, gliders, or boats). The resin is combined well with standard reinforcements (glass, carbon, aramid fibers). Epoxy resin LG 285 is certified by the German Luftfahrt-Bundesamtal. According to the information provided by the manufacturer Havel Composites (Přáslavice, Czech Republic), when heated at temperatures of 50 to 55 °C, it meets the application requirements in the design of gliders (operating temperature of −60 to +54 °C), and after heating at a temperature of 80 °C for 24 h, it begins to meet the aircraft requirements (operating temperature of −60 to +72 °C) [39]. Currently, this is one of the most commonly used aviation resins. Carbon fabric, 280 g/m^2^ in weight (ISO 4605) was characterized by a weave pattern 2 × 2 Twill (ISO 2113) and supersaturation of 220 g/m^2^. The volume fraction of fiber in the composites was on average 52%.

Ten-layer composites were produced by the pressing method, using the pressure of 2.5 MPa on a hydraulic press PDM—50S Mecamaq (Mecamaq, Lleida, Spain). The composite sheets were left for seven days to cure. Then, the samples (sized 100 mm × 15 mm) were prepared for testing. In order to give the cut composite appropriate hardness and resistance to a higher temperature, it was decided that the prepared sample must undergo heating, as recommended by the manufacturer. The process was executed in accordance with the instructions of the resin manufacturer (i.e., at 80 °C, for a period of twenty-four hours in the WKL 64/40 Weisstechnik chamber (Weiss Umwetltechnik GmbH, Heuchelheim, Germany)). The manufactured samples were exposed to different environmental operating factors such as UV-A radiation with high temperature and humidity as well as temperature shocks. The choice of environmental factors was determined by real environmental loads that affect aircraft skin during operations. The manufactured samples were divided into seven groups. Three groups were placed in an apparatus simulating adverse weather conditions (material weathering at a higher temperature and humidity) for a period successively corresponding to two, four, and six months of use in conditions that may occur during the hottest months of the year in a moderate climate [40,41,42]. Another three groups were exposed to thermal shocks, simulating dynamic changes in temperature, and corresponding to temperatures that occurred during flights of one hour over a period of six months. One group, which was treated as the control, was not exposed to any environmental loads.

### 2.2. Accelerated UV-A Weathering

In accelerated examinations, the authors used climatic data of the spring and summer seasons due to the occurrence of the highest temperature values and precipitation, which may adversely affect the structure of the composite. A simulation of the six-month exploitation time was conducted in accordance with EN ISO 4892-1 and PN-EN ISO 4892-3 standards. For this purpose, the accelerated weathering tester, UV QUV/SPRAY/RP Q-Lab Corporation, was used. The light source was the UV-A 340, simulating daylight [19]. The value of the intensity of radiation was 0.83 in/m^2^, the weathering time equaled 500 h, the length of the exposure time was 4 h, the length of condensation of water vapor cycle equaled 4 h, and the temperatures during exposure and condensation were 60 °C and 50 °C, respectively. The parameters (Table 2) were selected on the basis of experiences and observations included in research [43,44]. The time of accelerated weathering, corresponding to two, four, and six months of exposure to climatic conditions was successively 167 h, 334 h, and 500 h, respectively, of the weathering tester operation.

### 2.3. Thermal Shocks

The temperature shock test chamber, Shock Event T/60/V2 Weisstechnik (Weiss Umwetltechnik GmbH, Heuchelheim, Germany), was used to simulate changing loads associated with a sudden change in temperature. The examined material was exposed to thermal shocks. The values corresponding to the parameters of research are listed in Table 3.

### 2.4. Abrasive Wear

The very process of abrasion was executed on the basis of a linear tribometer, the Taber Linear Abraser (North Tonawanda, NY, USA). Half of the samples, both subjected to the impact of weathering and thermal shocks, underwent abrasive wearing without direct involvement of external factors in the process of friction. The other half underwent abrading in the presence of water. Water was introduced into the research due to its ability to significantly accelerate the process of tribological damage to composites, which results from the conducted preliminary testing and from the literature analysis (e.g., [3,45,46,47]). The tribometer, used during the tests described in this article, allowed for a sound simulation of the impact of different types of an abradant on the material surface. The samples were subjected to linear abrasion (Figure 1), caused by an abrasive element in an reciprocating movement, imitating the most common manner of an abrasive impact occurring in nature [48].

The samples of 105 mm × 15 mm, 3 mm thick were successively put into the working part of the tribometer. Next, a loaded tribometer pin was lowered onto it. The pin was ended with an abrasive material, 6.5 mm in diameter and 200 grade. The pin was pressed to the sample through a load of 1850 g. Water, in certain examinations, was introduced cyclically using a sprinkler. The frequency of movement of an anti-sample in the sample was 1.25 Hz. During one cycle of the reciprocating motion, the length of the friction path equaled 177.8 mm.

In the assumed intervals (every 0, 100, 300, 600, and 1000 abrasion cycles), the visual status, loss of weight were analyzed. The initial weight had been measured each time after each interval of the UV aging and thermal shocks. Surface maps of the material were created using precision scales, a profilometer, and an optical microscope. After each friction attempt, the sample was thoroughly cleaned each time with a brush or dried with paper. The abrasive wear and an assessment of the degree of the top layer wear were evaluated on the basis of weight loss with the use of precision scales XSE205DU/M (Mettler Toledo, Zurich, Switzerland), a visual condition using a microscope GX53 (Olympus Corporation, Tokyo, Japan), a camera No. SC180 (Olympus Corporation, Tokyo, Japan), as well as the surface topography of the samples made with the optical surface profilometer (MicroProf 100 FRT, FRT GmbH, Bergisch Gladbach, Germany). Surface maps were obtained using FRT Mark III software (III, FRT GmbH, Bergisch Gladbach, Germany) and the scanning system with an accuracy of 30 nm in the vertical axis and 100 nm in the horizontal axes. The measurement of the roughness parameter Rmax and the Ra surface parameter was made in accordance with the norm PN-EN ISO 4287 [49]. Maximum roughness depth Rmax means the highest profile peak and the depth of the deepest profile valley within the evaluation length based on the Rz parameter. Rz is a roughness depth and is the average value of the absolute values of the heights of the five highest profile peaks and the depths of the five deepest valleys within the evaluation length. The Ra arithmetical mean roughness value was calculated using the following relationship:(1)$$R_{a}=\frac{1}{l}\overset{l}{\underset{0}{\int }}\left|Z\left(x\right)\right|dx$$
where *l* is the length of the elementary segment [m]; and *Z* is the height of profile elements [m].

It was then decided to analyze changes in the composite hardness with regard to weathering and thermal shocks, since there is a link between tribological resistance and material hardness [50]. Therefore, the authors examined the hardness of the samples using the Shore D method by means of the hardness tester Digi Test II, Type DTAA Bareiss (FRT GmbH, Bergisch Gladbach, Germany), in accordance with the norm ISO 868.

In order to minimize the occurrence of structural defects, the greatest possible care was made in the production process of the composite. However, the possible changes on the surface of the abrasive stone, emerging as a result of a material wear, were minimized by recovering its surface every 100 abrasion cycles, in accordance with the manufacturer’s recommendations.

## 3. Tests Results

### 3.1. Evaluation of Abrasive Wear under the Influence of UV-A 

On the surface of the carbon-reinforced composites subjected to UV-A radiation, significant visual changes were observed, as illustrated in Figure 2. After the first interval, corresponding to two months of the impact of UV-A radiation in a natural environment, the photodegradation of the composite base was conspicuous, resulting in a change of the resin color from transparent to milky white. The next time interval demonstrated a deepened resin degradation and a local uncovering of carbon fibers of the first fabric layer, which after six months of the UV-A impact, were completely unprotected by a layer of epoxy resin.

By means of an optical microscope, the authors attempted to capture the impact of weathering on the composite surface that was not exposed to abrasive wear. Due to a large number of photos of the examined materials, the authors decided to present only the most relevant and most interesting pictures. In Figure 3a, the outer surface of the composite can be seen immediately after its manufacture. A smooth layer of resin was covered with imprints derived from the imperfect surface of the PET (polyethylene terephthalate) foil that was used for resin cross-linking to protect the mold plates from gluing. The fibers were difficult to notice and the number of defects (1) on the surface was low.

The impact of two months of atmospheric degradation is depicted in Figure 3b. Special attention needs to be drawn to a largely increased number of surface imperfections and defects. There were numerous cracks (2) and empty spaces (3). The carbon fibers of the fabric, hidden under a layer of the matrix base, began to emerge (4). Although the resin becomes destroyed as a result of photodegradation, it still constitutes a protective layer of the composite carbon reinforcement layers.

After four months of aging (Figure 3c), the first layer of carbon fabric that reinforces the composite becomes clearly visible. The remains of the degraded matrix can be observed, although it does not have a dominant share in the sample’s surface. The fibers themselves do not become visibly damaged in the applied method of observation and remain compact and undamaged. Minor impurities found in the carbon fabric become visible (5).

Another two months of UV-A impact caused further resin degradation (Figure 3d). Furthermore, damaged fibers began to appear. The amount of surrounding matrix was significantly depleted.

Figure 4 shows changes in the surface texture under the influence of UV-A. The surface texture was obtained by means of a profilometer.

After one hundred friction cycles, destruction of the matrix base became conspicuous. Additionally, the outer layer of the composite surface was completely scrapped. Nonetheless, significant damage to the fibers appeared only after 300 cycles. After the next interval, the material began to be ripped off and there was deeper damage. After 1000 cycles, the first layer of the composite reinforcement was removed in various sections. The picture of the surface abrasive wear is presented in Figure 5, with the example of a composite that was unaffected by adverse environmental conditions. After 100 cycles of friction, traces of abradant and slight exposures of fibers were noticeable, which in Figure 5a are arranged perpendicular to the direction of friction. Three hundred cycles of friction exposed the fibers of the first layer of carbon fabric (Figure 5b). The next three hundred cycles (Figure 5c) resulted in the formation of resin losses, which revealed the next layer of the fabric. The further process of friction caused the losses to enlarge (Figure 5d) as well as their number on the sample surface. Similar surface destruction was also observed in composites exposed to UV-A weathering and thermal shocks. Moreover, Figure 6 and Figure 7 show sample maps of the surface topography of the composite surface after consecutive intervals of the friction process for dry and wet friction, respectively, after a six-month impact of adverse external conditions. An observation of the maps allowed us to determine that during wet abrasion, the dent of the anti-sample was significantly higher than in the case of dry abrasion. The walls of the dent were smoothed, and the bottom of the friction path was less smooth than in dry friction.

An observation of the material surface after consecutive intervals in the friction process indicates a typically abrasive nature of abrasive wear, regardless of the time of the UV-A impact or thermal shocks.

Abrasive wear with water is characterized by a significantly larger scatter of individual measurements than in the case of dry friction (Figure 8), which also translates into the size of the standard error of the mean depicted in the frame (Figure 9). Approximation functions were determined by the least-squares method (Figure 8) as indicated by the coefficients “a” in the linear function y = ax + b, and were higher in the case of wet friction. UV-A and thermal shocks influenced the nature of intensity of tearing off subsequent material particles in successive friction cycles. An increase in intensity implies that along with the progress of the friction process, larger lumps of the material were detached. Under the wet condition in Figure 8, the slope of the regression line of the data increased as the number of months of aging changed from 0 to 2 to 4, but the distribution of the data converged on the low weight loss side. Therefore, regarding the effect of UV exposure, although rapid progress of local abrasive wear may be induced by accelerated deterioration of the material, it is considered that change of material properties improves overall wear durability.

As for the principle, the biggest and most dynamic weight loss occurred in the case of composites that were not exposed to the influence of UV-A (Figure 9a) and subjected to friction with water. After 1000 cycles, the loss of weight was on average 0.235 g, more than 110 times than during dry abrasion. After successive months of weathering, there was a considerable increase in resistance to abrasion (Figure 9b–d) to a level close to the damage that appeared in dry friction (i.e., to the average value of 0.059 g). Moreover, the semi-annual exposure to UV-A radiation meant that there was a reduction in the consumption of abrasive material. The weight loss decreased with the number of months of aging, and it was considered that overall wear durability was improved as in the case of Figure 8. On this basis, it can be concluded that the impact of UV-A for a period of up to six months had a positive influence on composite resistance to abrasive wear during the process of friction in the water environment, and adversely during dry friction.

The average values of the Ra roughness parameter after the process of friction for various time intervals of UV-A impact are juxtaposed in Figure 10. It can be observed that in accordance with the expected principle, the increase in the number of friction cycles resulted in the rise in Ra parameter. The changes were significant as confirmed by an analysis of the Mann–Whitney U tests. Surface roughness was greater and the nature of the changes was more intensive in the case of friction in direct contact with water. Therefore, water is not a lubricant. The 3D graph of the Ra response parameter (the actual findings are marked with blue points) to a change in external factors was performed using the least-squares method. This was similar in the case of the Rmax parameter values (Figure 11). Changing the values of both roughness parameters, both in relation to the UV-A impact time and the nature of friction, corresponds to the results of the loss of weight.

### 3.2. Impact of Thermal Shocks on Abrasive Wear

By means of an optical microscope, the authors attempted to capture surface changes due to the influence of thermal shocks and tribological wear. In comparison with the image shown in Figure 12a (composite immediately after its manufacture with visible slight lack of resin continuity (1)), in the image shown in Figure 12b, it is possible to observe more numerous imperfections in the surface layer of the resin in the form of small pits stacked along from the imperfect surface of the PET foil used during pressing (2) and the law of revealing cavities that uncovered the reinforcement (3). Further cycles of thermal shocks (simulating the duration of such a phenomenon for another two months) caused more destruction to the surface layer. There were many pits, streamlined in shape (4). The six-month impact of thermal shocks caused clear losses of the resin, leading to extensive uncovering of the reinforcement area. Similarly, Figure 13 shows changes in the surface texture of the composite exposed to thermal shocks. The surface maps of these samples after dry and wet friction were visually similar to those undergoing UV-A weathering (Figure 6 and Figure 7).

An analysis of the results of the abrasive wear under the influence of thermal shocks was carried out in an analogical way as under UV-A radiation. Figure 14 juxtaposes the characteristics of the intensity of the composite wear in the process of dry friction and in the presence of water. In relation to the process of friction with regard to the effects of a harmful environment of thermal shocks, the intensity of abrasive wear during dry friction, along with its further progress, was slightly decreased (the coefficients “a” of the regression function decreased). Thus, the effect of thermal shocks in the case of resistance to weight loss during the friction process also turned out to be favorable. In the case of friction with water, the abrasive wear intensity remained at a similar level and the friction process had stabilized. Similarly, and in the case of environmental thermal shocks, it is possible to observe a significant influence of water in the process of friction in the size of the standard mean error, spread of extreme results, and the size of the weight loss (Figure 15). Similarly, there was a positive impact of a long-term effect of thermal shocks on the improvement of material properties in terms of reducing abrasive wear. 

Analyzing the intensity of abrasive wear in the case of dry friction with the increase in the exposure time of both UV-A (Figure 8 and Figure 9) and thermal shock (Figure 14 and Figure 15), a significant similarity like changes was noticeable. In both cases, the loss of mass was insignificant and did not increase significantly with the time of environmental impact. The weight loss after 1000 friction cycles did not exceed 0.05 g. The repeatability of the measurements was high. The spread of the results was small. On the other hand, in the case of wet friction, the environmental impact was very large. In both cases, they will have a positive effect on changes in the abrasive wear properties of the material. The result was the same after six months. After 1000 cycles of friction, the average weight loss was about 0.06 g. Nevertheless, the effect was slightly worse after two months of exposure to heat shocks. In the case of UV-A interaction, the changes in material properties took place gradually. Changes in the structure of the material require further physico-chemical analysis.

Thermal shocks, in contrast to UV-A, except for the first period of a two-month impact, caused a significant decrease in the Ra parameter to a lower level than before exposing the composites to changes in external conditions (Figure 16 and Figure 17). An environment of thermal shocks influenced the reduction in the differences in Ra roughness values between dry friction and friction in the presence of water. Figure 16 and Figure 17 show a spatial diagram of the effect of time of thermal shocks and the number of cycles on the roughness parameters Ra and Rmax. The characteristics of changes in both parameters was similar. Both in the first and the second parameter, the lowest values were reached after a six-month period of thermal shocks, whereas the highest one was after two months. It can be said that thermal shocks have a stabilizing effect to the response of a composite to the friction process.

### 3.3. Others Results

The experimental results were developed statistically. The basic statistical analysis was made, adjusting the distribution in terms of skewness and kurtosis. The results of adjusting a normal distribution were not in the confidence interval. The influence of the environment on the loss of mass and surface roughness was also assessed, taking into consideration the scope of the measurement errors. Bearing in mind the fact that skewness and kurtosis tremendously differ from the zero value, exceeding unity, in order to determine a significant influence, a non-parametric test was used with regard to two independent groups, the extremely sensitive Mann–Whitney U test. The *U* statistics with the number of attempts *n* ≤ 20 is expressed by the following equation [51],
(2)$$U\left(n_{1},n_{2}\right)=Smaller\;of\left[\begin{array}{c} n_{1}n_{2}+\frac{n_{1}\left(n_{1}+1\right)}{2}-\sum^{\;}R_{1}\\ n_{1}n_{2}+\frac{n_{2}\left(n_{2}+1\right)}{2}-\sum^{\;}R_{2} \end{array}\right]$$
where Ʃ *R*_1_ and Ʃ *R*_2_ are the rank sums assigned to individual results in both sample groups, counting *n*_1_ and *n*_2_. The *U* statistics should comply with the following equation:(3)$$n_{1}n_{2}=U_{1}+U_{2}$$

Additionally, the relationship should be fulfilled:(4)$$\sum^{\;}R_{1}+\sum^{\;}R_{2}=\frac{N\left(N+1\right)}{2}\;where\;N=n_{1}+n_{2}$$

In order to fulfil the hypothesis of the statistically significant change, *U* statistics must be below the critical values than the critical value *U_kr_* read from the tables, appropriate to the adopted level of confidence *α* (usually *α* < 0.05) and the number of groups *n*_1_ and *n*_2_. Number of groups was 10.

The above-mentioned changes regarding the influence of UV-A radiation and thermal shocks were significant in their character, as confirmed by nonparametric tests in relation to the two independent groups. The vast majority of the assessment results (Table 4) of the statistical significance of the impact of time of thermal shocks and UV-A radiation on weight loss, depending on the type and number of friction cycles with regard to the initial state, were significant by a probability *p* < 0.05. Similarly, as indicated by previously presented research findings, the significance was confirmed in relation to the parameter Ra. For this reason, none of the variable factors (i.e., time of impact of adverse external conditions or the type of friction) will be excluded in further stages of the research.

The hardness of the composite material did not change significantly within four months of UV-A impact (Figure 18). However, after the simulation of the six-month exposure to UV-A radiation and thermal shocks, there was a significant increase in hardness, which is probably related to progressive additional cross-linking of the composite matrix base. It was also noticeable that the composite hardness was greater when it was exposed to thermal shocks. It was also possible to observe an analogy of weight loss in the process of friction with water to hardened material. Along with greater hardness, the material was less susceptible to wear in the event of friction in the presence of water. Therefore, this tendency is correct, as in the case of materials on the basis of metal elements, where higher hardness goes hand in hand with higher resistance to wear [52,53] and is in agreement with certain polymer materials that are in common use [4]. However, such a relationship with regard to dry friction was not present.

## 4. Conclusions

UV-A weathering largely influenced the degradation of the outer layer of the composite matrix base, leading to its excessive damage. Damage to the composite matrix base was much higher than the damage caused by cyclic changes as a result of changes in cyclic temperature values. The biggest changes in the composite surface roughness in the case of exposure to thermal shocks occurred in the initial phase of abrasive wear when the outer layer of the material was damaged. In general, it was found that the two external environments contributed to damaging the composite surface layer, in particular the base matrix, uncovering the reinforcement. Weight loss of composites after six months of exposure to thermal shocks in wet friction was on average 3.4 times higher than in dry friction. However, in the case of composites subjected to UV-A treatment, it was on average 1.5 times higher.

Overall, the introduction of water between a cutting (friction) pair in the process of friction significantly increased the tribological loss of the examined material. Nevertheless, the impact of UV-A and thermal shocks throughout the examined period favorably affected the composite resistance to the abrasive wear of the material during the process of friction in a water environment. Over a 100-fold reduction in weight loss was observed. This indicates that far-reaching thermal degradation of the polymer has not yet occurred in the studied temperature and time range, which would be the cause of rapid deterioration in abrasive wear resistance. Changes in the structure of the material after the impact of UV-A and thermal shocks require further physico-chemical analysis. Reduction in weight loss indicates the need to modify the principles of a technological preparation of composites from the examined resin and to consider an extension of the recommended time for resin heating.

At the same time, it was noticed that both rapid temperature changes and UV-A radiation combined with increased relative humidity and ambient temperature, led to the deterioration of resistance to dry abrasive wear by approximately 50%.

The research findings, presented in this article, do not clearly show the dependence of tribological resistance of the epoxy carbon-reinforced composite upon its hardness. The composite hardness after six months of thermal shocks increased by 2.6% and after UV-A, it increased by 4.2% and on average amounted to 90.8 and 92.3° Sh, respectively. In the case of the friction processes in a water environment, a rise in material hardness increased resistance to abrasive wear, and in the case of dry friction, it impaired the resistance.

The above research, resulting from the need to determine the behavior after an assumed exploitation time, in unfavorable conditions, was confirmed by the operating data of composite materials used in aviation. The assessment of the exploitation condition of a material allows for the verification of the safety factors and recommendations of use.

## Figures and Tables

**Figure 1 materials-13-03965-f001:**
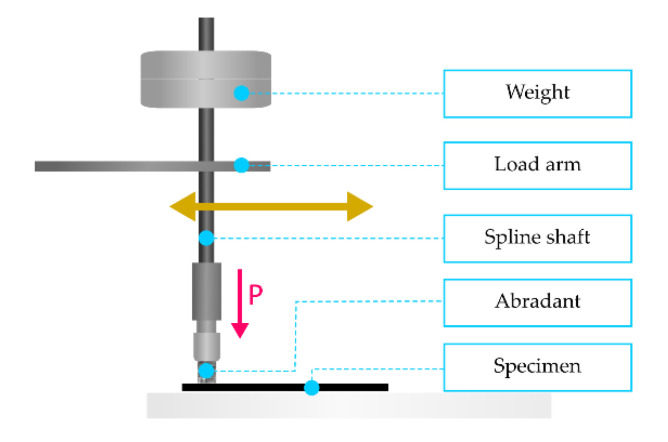
Scheme of the action of force during the friction process.

**Figure 2 materials-13-03965-f002:**
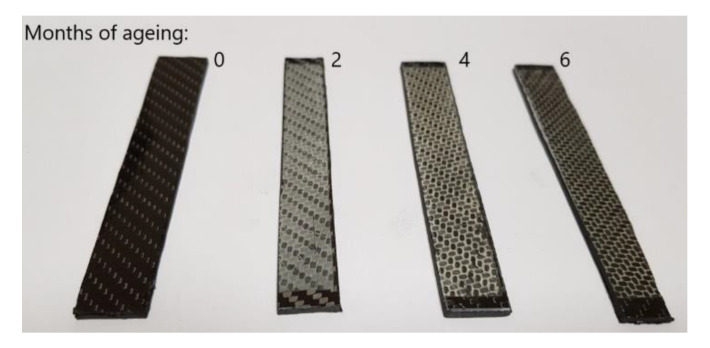
Condition of visual samples after the process of accelerated weathering. From the left, the samples corresponding to 0, 2, 4, and 6 months of use.

**Figure 3 materials-13-03965-f003:**
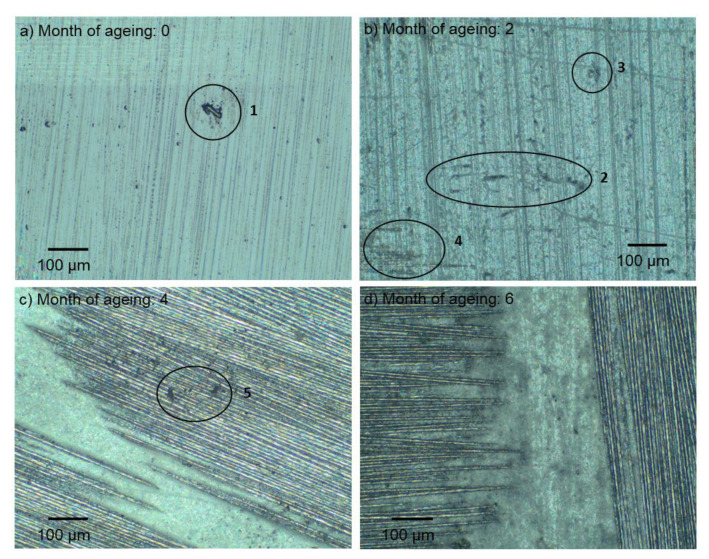
Microscopic image of the examined sample surface prior to abrasive wear after (**a**) 0, (**b**) 2, (**c**) 4, and (**d**) 6 months of weathering.

**Figure 4 materials-13-03965-f004:**
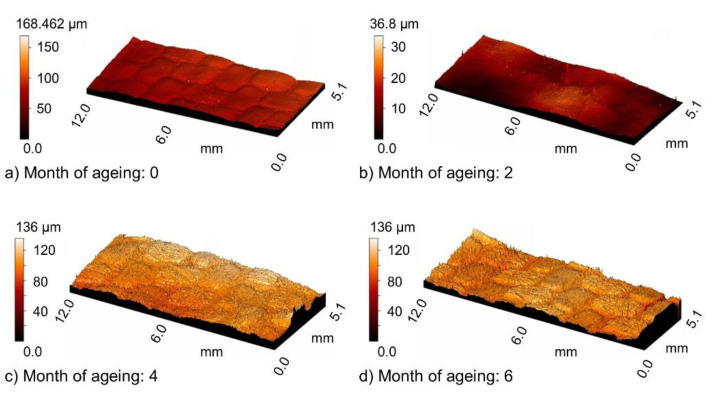
Texture of the examined surface prior to abrasive wear after (**a**) 0, (**b**) 2, (**c**) 4, and (**d**) 6 months of weathering.

**Figure 5 materials-13-03965-f005:**
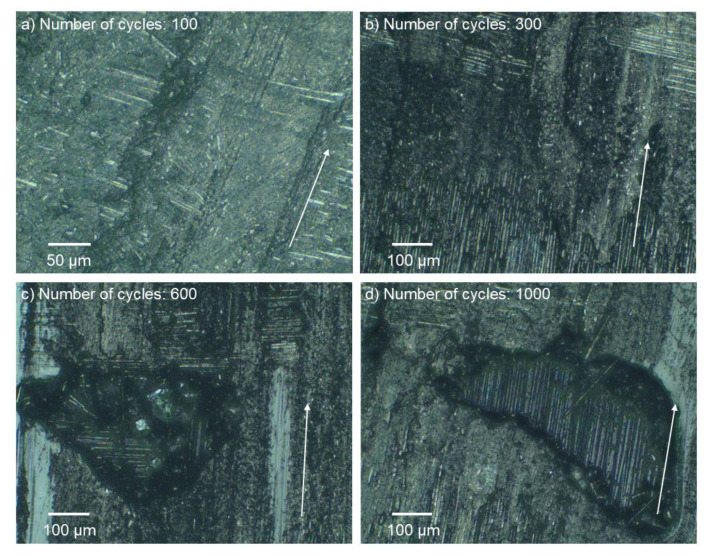
Changes on the composite surface that did not undergo weathering after: (**a**) 100, (**b**) 300, (**c**) 600, and (**d**) 1000 cycles of dry abrasive wear (arrows show the direction of the reciprocating sliding).

**Figure 6 materials-13-03965-f006:**
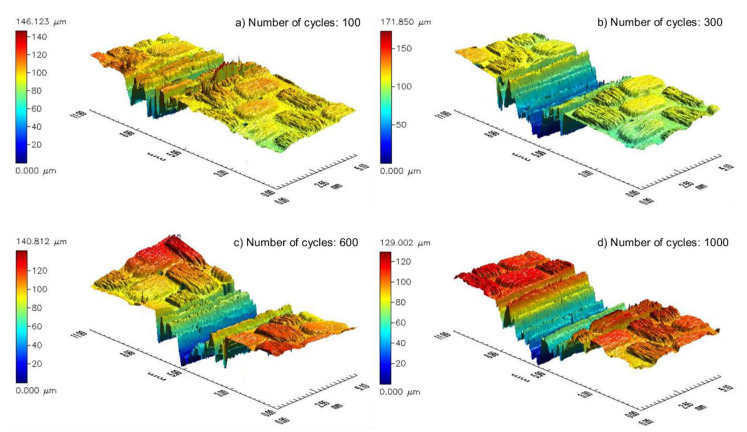
Exemplary maps of the surface sample (six months of weathering) after (**a**) 100, (**b**) 300, (**c**) 600, and (**d**) 1000 cycles of dry abrasive wear.

**Figure 7 materials-13-03965-f007:**
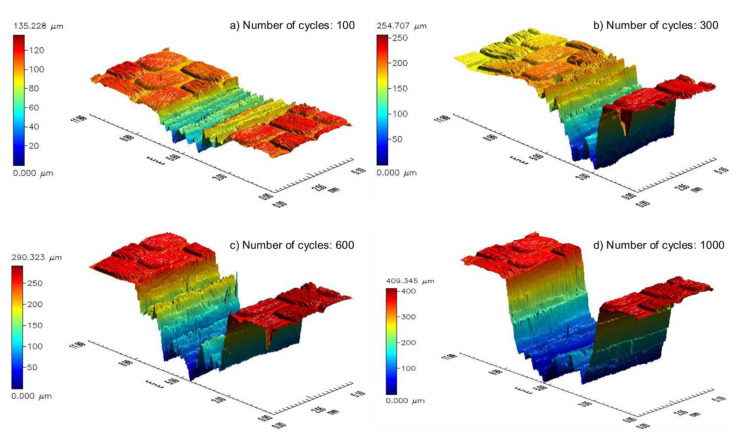
Exemplary maps of the surface sample (six months of weathering) after (**a**) 100, (**b**) 300, (**c**) 600, and (**d**) 1000 cycles of wet abrasive wear.

**Figure 8 materials-13-03965-f008:**
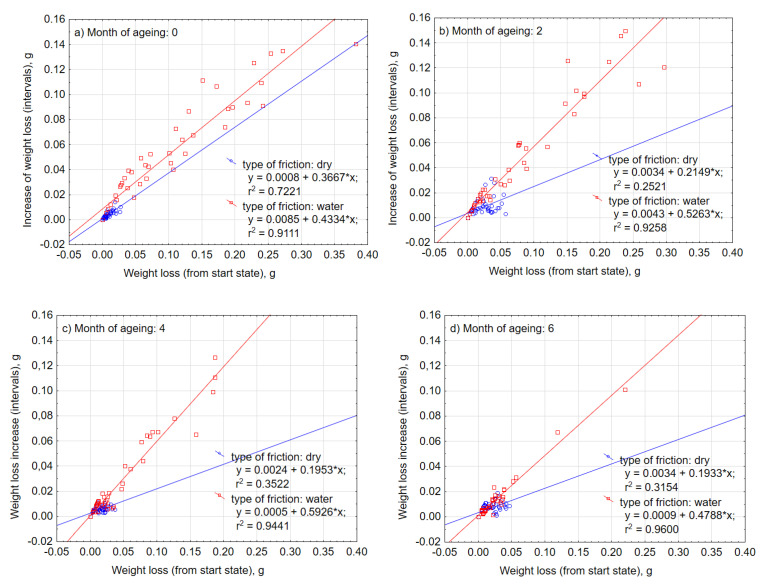
Scattering of the measurements of composite weight loss after (**a**) 0, (**b**) 2, (**c**) 4, and (**d**) 6 months of UV-A impact.

**Figure 9 materials-13-03965-f009:**
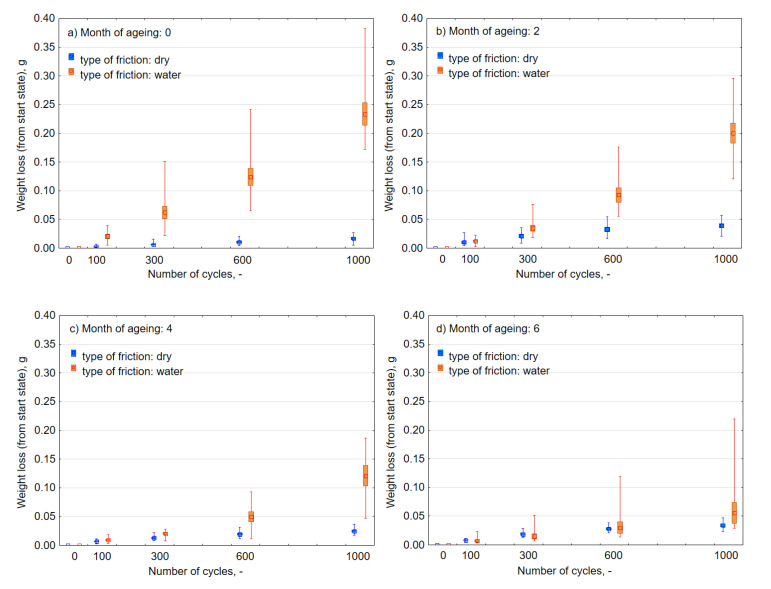
Average loss of the weight of samples that underwent weathering after the assumed number of cycles of dry and wet abrasion (**a**) 0, (**b**) 2, (**c**) 4, and (**d**) 6 months of UV-A impact.

**Figure 10 materials-13-03965-f010:**
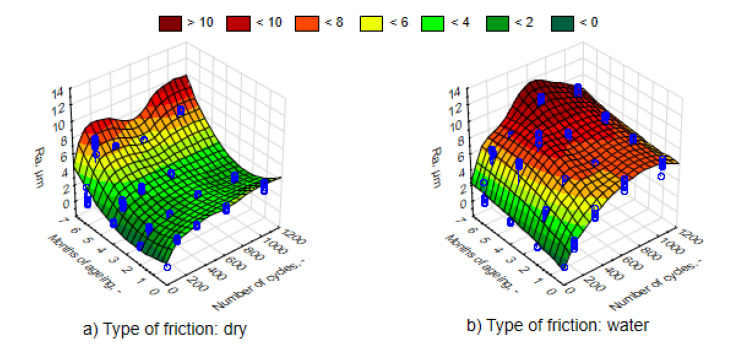
Characteristics of changes in parameter Ra, depending on the number of friction cycles, for samples in different phases of weathering, (**a**) dry friction, (**b**) wet friction.

**Figure 11 materials-13-03965-f011:**
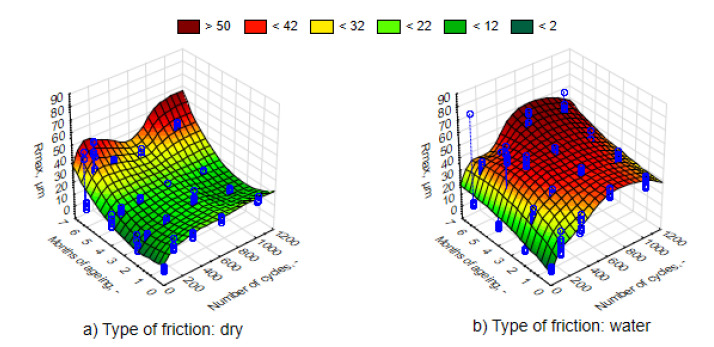
Characteristics of changes in parameter Rmax, depending on the number of dry friction cycles, for samples in different phases of weathering, (**a**) dry friction, (**b**) wet friction.

**Figure 12 materials-13-03965-f012:**
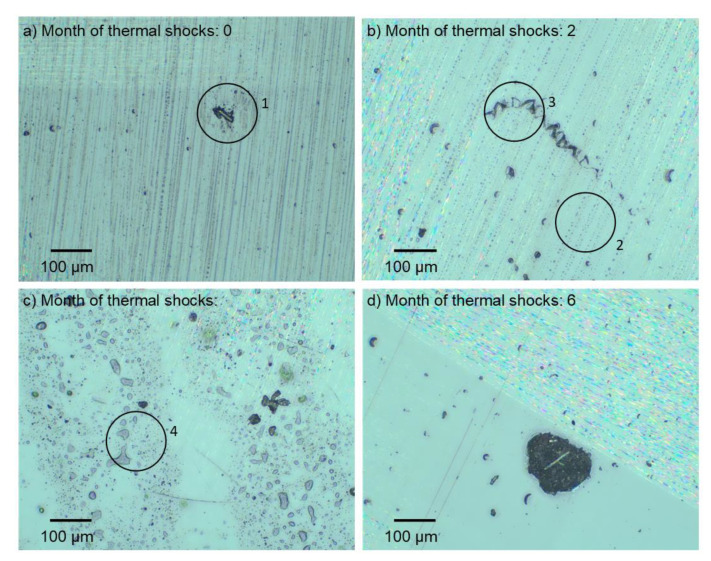
Surface of the examined sample prior to abrasive wear after (**a**) 0, (**b**) 2, (**c**) 4, and (**d**) 6 months of thermal shocks.

**Figure 13 materials-13-03965-f013:**
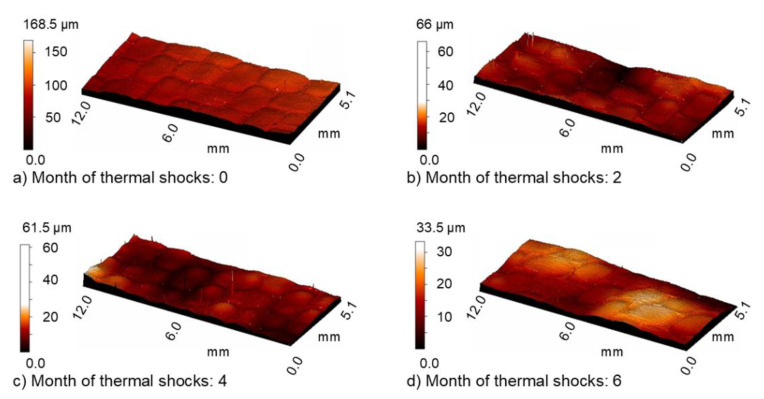
Surface of the examined sample prior to abrasive wear after (**a**) 0, (**b**) 2, (**c**) 4, and (**d**) 6 months of thermal shocks.

**Figure 14 materials-13-03965-f014:**
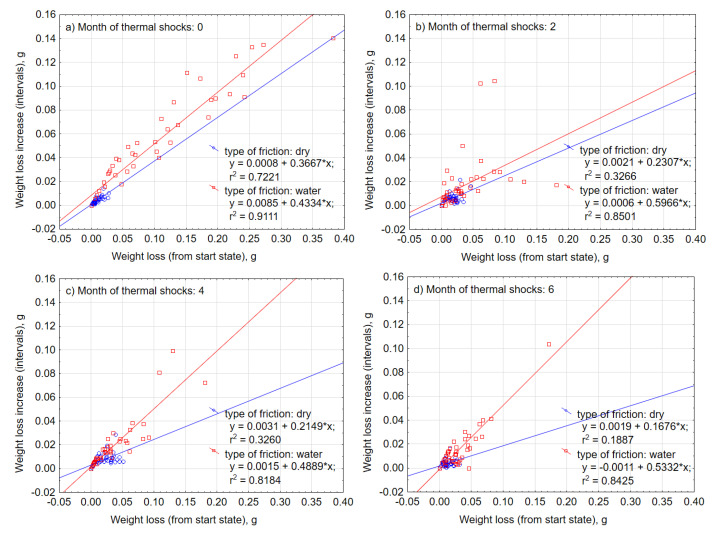
Scattering of the measurements of weight loss after the impact of (**a**) 0, (**b**) 2, (**c**) 4, and (**d**) 6 months of thermal shocks.

**Figure 15 materials-13-03965-f015:**
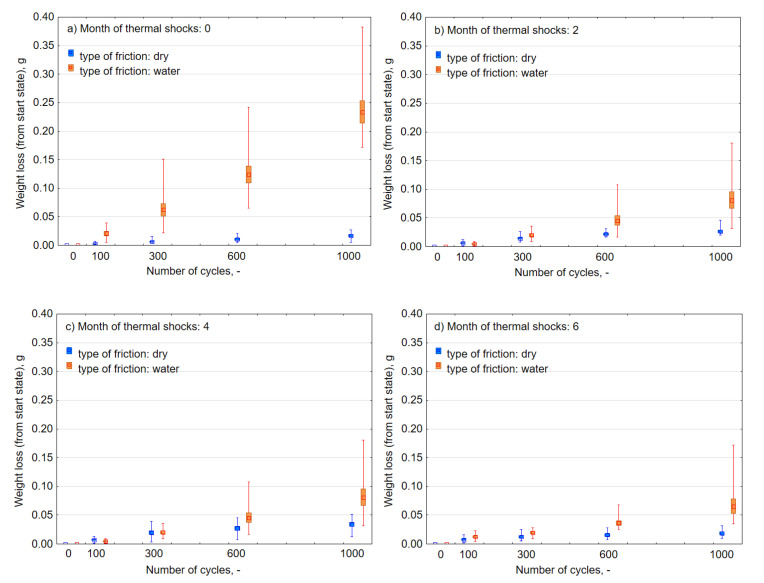
Average weight loss of individual samples subjected to thermal shocks, after the assumed number of cycles of dry and wet abrasion, (**a**) 0, (**b**) 2, (**c**) 4, and (**d**) 6 months of thermal schocks.

**Figure 16 materials-13-03965-f016:**
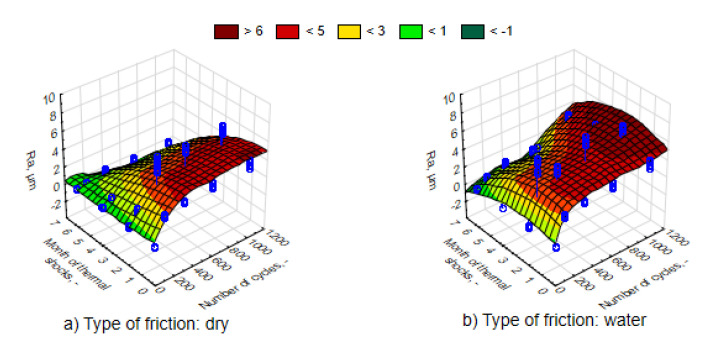
Characteristics of changes in the parameter Ra, depending on the number of friction cycles, for samples exposed to thermal shocks, (**a**) dry friction, (**b**) wet friction.

**Figure 17 materials-13-03965-f017:**
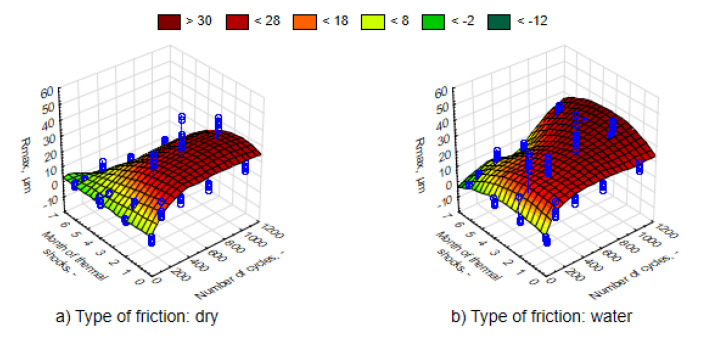
Characteristics of changes in the parameter Rmax, depending on the number of friction cycles, for samples exposed to thermal shocks, (**a**) dry friction, (**b**) wet friction.

**Figure 18 materials-13-03965-f018:**
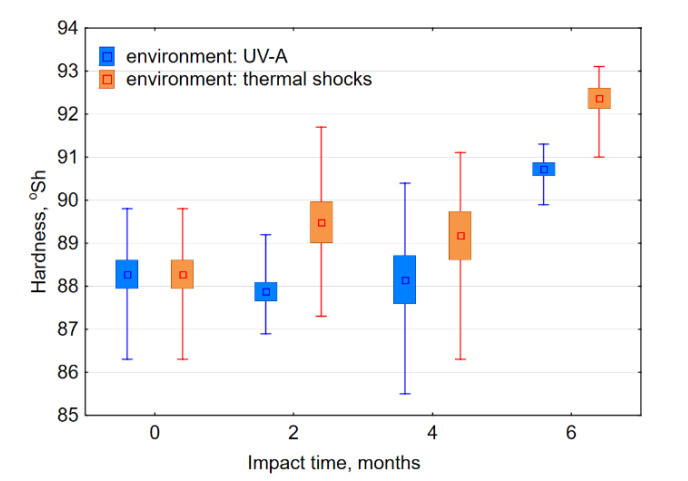
Average hardness of the samples after assumed weathering periods. The error bars mark the values of average standard deviation values of particular data series.

**Table 1 materials-13-03965-t001:** Summary of the conducted research.

Examined Scope	Examples	Conclusions	Bibliography
Composition of polymer composites	The presence of inorganic particles (e.g., Al_2_O_3_, TiO_2_, ZnO, CuO, SiC, ZrO_2_, Si_3_N_4,_ SiO_2_, CaCO_3_)	The possibility of influencing the tribological properties of polymer composites after introducing physical modifiers into their composition was observed.	[12,13,14,15,16,17,18,19]
	Various allotropic forms of the element (e.g., carbon-graphite, graphene)	Different tribological properties of materials containing additives of various allotoropic form were observed.	[12,16,17,18,19,20,21,22]
	Shaping tribological properties with particles of various sizes	Observation of clear differences between polymer composites with the presence of physical friction modifiers characterized by different grain sizes.	[23,24,25,26,27]
	Type/manner of laying the fabric	The authors characterize mechanical and tribological properties obtained for composites made of various fabric and various manner of laying.	[28,29,30,31,32]
Analyze of impact of composite manufacturing conditions	-	A change in the properties of polymer composites depending on the technology used to manufacture them was observed. The existing impact should be determined each time.	[33,34]
Impact of external factors between flights	-	As the exposure to environmental conditions increased, the composite material was more affected.	[35]

**Table 2 materials-13-03965-t002:** Research assumptions of accelerated UV-A weathering.

The Value of the Intensity of Radiation	The Weathering Time	The Length of the Exposure Time	The Length of Condensation of Water Vapor Cycle	The Temperatures during Exposure	Time of the Weathering Tester Operation
0.83 in/m^2^	500 h	4 h	4 h	50 °C/60 °C	167 h/334 h/500 h

**Table 3 materials-13-03965-t003:** Research assumptions of thermal shocks.

Factor	Adopted Values
The flight time	1 h
Number of flights per day	2
Preparation time of the aircraft in-between flights	0.5 h
Time of the aircraft remaining at a ceiling of 11 km	0.5 h
Temperature of the aircraft skin on the ground	+60 °C
Temperature of the aircraft skin at a ceiling of 11 km	−56.5 °C
Number of cycles of temperature changes	728
Temperature difference	116.5 °C

**Table 4 materials-13-03965-t004:** Assessment results of the statistical significance of the impact of time of environment on the weight loss, depending on the type and number of friction cycles (the impact is significant if probability *p* < 0.05).

Time of Impact of Environment	Type of Friction	Number of Friction Cycles	UV-A		Thermal Shocks	
			U	p	U	p
2	dry	100	5.0	0.0008	14.0	0.0073
		300	4.0	0.0006	8.0	0.0017
		600	3.0	0.0004	7.0	0.0013
		1000	2.0	0.0003	18.0	0.0172
	with water	100	29.0	0.1212	9.0	0.0022
		300	20.0	0.0257	5.0	0.0008
		600	23.0	0.0452	5.0	0.0008
		1000	34.0	0.2413	1.0	0.0002
4	dry	100	7.0	0.0013	12.0	0.0046
		300	7.0	0.0013	11.0	0.0036
		600	12.0	0.0046	9.0	0.0022
		1000	19.0	0.0211	10.0	0.0028
	with water	100	21.0	0.0312	9.0	0.0022
		300	5.0	0.0008	5.0	0.0007
		600	3.0	0.0004	5.0	0.0007
		1000	5.0	0.0008	1.0	0.0002
6	dry	100	3.0	0.0004	7.0	0.0013
		300	4.0	0.00063	16.0	0.0113
		600	0.0	0.0002	26.0	0.0757
		1000	4.0	0.0006	45.0	0.7337
	with water	100	9.0	0.0022	31.0	0.1620
		300	4.0	0.0006	5.0	0.0008
		600	5.0	0.0008	1.0	0.0002
		1000	5.0	0.0008	0.0	0.0002

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
