# Peer review of "The Impact of Selected Atmospheric Conditions on the Process of Abrasive Wear of CFRP"

_materials, 2020, doi:10.3390/ma13183965_

Round 1

Reviewer 1 Report

I have read this scientific paper with interest and I consider that it can be published with a small modification specified below.

Mark in figure 3 (on each micrograph) the defects described above in text. For example I attached a picture and must be complete with text:
1. The composite can be seen immediately after its manufacture with few defects ...
2. After two months of ageing, the first layer of carbon fabric .... and the defects ....
...

The same observation for figure 12.

Author Response

Thank you for your review.

According to your suggestions, the changes are included in the text and marked in red.

Best regards

Authors

Reviewer 2 Report

Comments: The work presented by the authors are very interesting. I have some major concerns about the quality of this article. Kindly, address the following issues:

General Comments:

  1. Title is very lengthy. I suggested to rewrite it in short form and make it more informative.
  2. Abstract: No need to provide more paragraphs. One single paragraph is perfect in abstract section.
  3. Introduction: The references are added unnecessary in the sentences. For instance: Page 2, line 63: what is the meaning of adding [12–19] references in one single line. It’s better to split and shorten the references.
  4. Section 2: Figure 1 does not show anything. It’s better to provide schematic diagram of complete paper.
  5. Conclusion: The conclusion needs some quantitative results. Please modify according to it.

Technical Comments:

  1. The technical part of this paper is weak. It’s better to compare the results and discussion with other previous published works.
  2. Section 3: It’s better to mention the full labelling on surface of the examined samples figures.
  3. Page 7, Line 238: Where is the Figure 2-0?
  4. The description of Figure 3 & 5 is very less. Please include some basic mechanism to clearly justify the mechanism.
  5. Captions of figures are not same everywhere. Please check it.

Author Response

Dear Reviewer,

Thank you for your review.

We focused on analyzing this review and made the appropriate changes. We are grateful for help in shaping the new form of our publication. Below are our explanations. All changes in the text are marked in red colour

Comment:

Comments: The work presented by the authors are very interesting. I have some major concerns about the quality of this article. Kindly, address the following issues:

General Comments:

  1. Title is very lengthy. I suggested to rewrite it in short form and make it more informative.

Response:

We propose the following title: The Impact of Selected Atmospheric Conditions on the Process of Abrasive Wear of CFRP.

Comment:

  1. Abstract: No need to provide more paragraphs. One single paragraph is perfect in abstract section.

Response:

Abstract has been shortened.

Comment:

  1. Introduction: The references are added unnecessary in the sentences. For instance: Page 2, line 63: what is the meaning of adding [12–19] references in one single line. It’s better to split and shorten the references.

Response:

We have made same corrections.

Comment:

  1. Section 2: Figure 1 does not show anything. It’s better to provide schematic diagram of complete paper.

Response:

Thank you. We rebuilt this Figure.

Comment:

  1. Conclusion: The conclusion needs some quantitative results. Please modify according to it.

Response:

We have modified a few sentences in the conclusions.

Comment:

Technical Comments:

  1. The technical part of this paper is weak. It’s better to compare the results and discussion with other previous published works.

Response:

The following text is included in the introduction:

„The analysis of the available literature regarding the abrasion of composite materials has allowed focusing on the possibilities of supplementing previously unexamined areas. Despite a lot of research in tribology of polymer composites, there are in literature, not a results they show of influence the thermal shocks as an environment on the tribology properties.”

While analyzing the literature, we did not come across any other scientific research that would discuss the results based on such a combination of research methods, especially regarding the long-term influence of changing weather conditions. A comparison to other results that are not similar in terms of content and technology could be inadequate. And it could raise concerns as to the legitimacy of comparing the results obtained with other research methods.

Comment:

  1. Section 3: It’s better to mention the full labelling on surface of the examined samples figures.

Response:

We have modified a labelling of Figures.

Comment:

  1. Page 7, Line 238: Where is the Figure 2-0?

Response:

We corrected this mistake. It is Figure 3-0. We changed the numbering to the standard one: 3a.

Comment:

  1. The description of Figure 3 & 5 is very less. Please include some basic mechanism to clearly justify the mechanism.

Response:

Figure 3 is described in the text above, starting from line 227. However, we have extended the description to Figure 5 with the following text:

„After 100 cycles of friction, traces of abradant and slight exposures of fibers are noticeable, which in Figure 5a are arranged perpendicular to the direction of friction. 300 cycles of friction exposed the fibers of the first layer of carbon fabric (Fig. 5b). The next three hundred cycles (Fig. 5c) resulted in the formation of resin losses, which revealed the next layer of the fabric. The further process of friction caused the losses to enlarge (Fig. 5d), as well as their number on the sample surface.”

Comment:

  1. Captions of figures are not same everywhere. Please check it.

Response:

We verified the descriptions of the Figures. We hope they will be properly described now.

Yours faithfully

Authors

Reviewer 3 Report

Thank you for your article to Materials.
Though your idea and approach are interesting, major revisions are required to publish this manuscript as a paper. Please consider the following comments.

(1) Only the tendency of experimental data is described in your manuscript, and there is no logical discussion of why it happened from the viewpoint of material science and tribological analysis.
It is necessary to consider how UV exposure and thermal shock affect the physical chemical condition and properties of materials and how water affects the wear phenomenon, and then to consider their combined influences to the abrasive wear reported in your manuscript.

(2) Rmax is an old surface roughness parameter. Rz, which represents the maximum height roughness in the current ISO, is thought to be more suitable for your manuscript. However, it is unclear why Ra and Rmax (Rz) are evaluated separately in your manuscript.
For example, is it possible to consider Ra and Rmax (Rz) in relation to damage of the epoxy resin matrix base and damage of the carbon fabric, respectively?

(3) The description concerning Figs. 8 and 14 is insufficient. Differences between the effects of UV exposure and thermal shock should be considered based on the slope and intercept of the regression line.

(4) Many mistakes such as the following are found in your manuscript. Please recheck the whole manuscript carefully.
Line 66: Unnecessary space in front of period
Line 93: No period
Line 133: Sign between 50°C and 55°C do not make sense
Line 186 and 190: Inappropriate reference numbers
Line 193(Caption of Figure 1): Inappropriate reference number
Line 202: Disunity among "aging" and "ageing"
Equation (1): Three same expressions are lined up
Line 218: Expression of "the a "
Line 231: No period
Line 247 and 252: "Fig. 3-5" and "Fig. 3-7" are difficult to distinguish from "Figs. 3-5" and "Figs. 3-7"
Figure 3: The scales of upper figures are hidden
Figure 4: Not sure which figure corresponds to which condition
Line 268: "Figure 4" seems to be a mistake in "Figure 5"
Figures 6 and 7: Not sure which figure corresponds to which condition
Line 286(Caption of Figure 6): "100, 300, 600 and 1,000 of dry abrasive wear" seems to be a mistake in "100, 300, 600 and 1,000 cycles of dry abrasive wear"
Line 290(Caption of Figure 7): "100, 300, 600 and 1,000 of wet abrasive wear" seems to be a mistake in "100, 300, 600 and 1,000 cycles of wet abrasive wear"
Line 322(Caption of Figure 11): "dry" is unnecessary
Figure 12: The scales of upper figures are hidden
Figure 13: Not sure which figure corresponds to which condition
Line 370(Caption of Figure 15): "dry abrasion" seems to be a mistake in "dry and wet abrasion"
Equation (2): "smaler" seems to be a mistake in "smaller"
Table 4: Inconsistency with text (Contains non-significant results)
Line 428(Caption of Figure 18): No period with first sentence
Line 430: No chapter number

Author Response

Dear Reviewer,

Thank you for your review. We focused on analyzing this review and made the appropriate changes. We are grateful for help in shaping the new form of our publication. Below are our explanations. All changes in the text are marked in red colour.

Comment:

Thank you for your article to Materials.

Though your idea and approach are interesting, major revisions are required to publish this manuscript as a paper. Please consider the following comments.

(1) Only the tendency of experimental data is described in your manuscript, and there is no logical discussion of why it happened from the viewpoint of material science and tribological analysis.

It is necessary to consider how UV exposure and thermal shock affect the physical chemical condition and properties of materials and how water affects the wear phenomenon, and then to consider their combined influences to the abrasive wear reported in your manuscript.

Response:

In our opinion, the article does not concern research related to polymer chemistry, therefore enriching the article with detailed information on the impact of heat shocks and UV-A on the physicochemical properties of materials (for example: DSC test results with interpretation) would change the nature of the work and significantly extend the scope of the research results.

Nevertheless, we agree with the editor's note that such research is important and should be checked. At this point, we want to inform you that the research is continuing. We already have most of the research results on mechanical properties. It is related to our testing program to establish performance recommendations for these composites. Additionally, we were considering performing chemical tests and checking, for example, the degree of resin cross-linking. After this comment, we can see that we should include them in the research agenda. More of our articles in this research area will be published soon.

Comment:

 (2) Rmax is an old surface roughness parameter. Rz, which represents the maximum height roughness in the current ISO, is thought to be more suitable for your manuscript. However, it is unclear why Ra and Rmax (Rz) are evaluated separately in your manuscript.

For example, is it possible to consider Ra and Rmax (Rz) in relation to damage of the epoxy resin matrix base and damage of the carbon fabric, respectively?

Response:

Parameter Rz in ISO means a sum of the height of the highest profile peak and the depth of the deepest profile valley, relative to the mean line, within a sampling length. In the case of the tested composites, the defects visible in Figure 5 were very irregular in terms of depth and also in terms of their frequency of occurrence on the friction surface. This resulted in very large discrepancies in the results of Rz. From a statistical point of view, the inference was unjustified. Therefore, we have decided to set the Rmax parameter, which is the highest profile peak and the depth of the deepest profile valley within the evaluation length. The scatter of the Rmax parameter results was much smaller. It was possible to use these results to prepare the characteristics of the tribological properties of the tested composites.

Comment:

 (3) The description concerning Figs. 8 and 14 is insufficient. Differences between the effects of UV exposure and thermal shock should be considered based on the slope and intercept of the regression line.

Response:

We have made slight modifications to the description of Figures 8 and 14. They should now be more appropriate.

Comment:

 (4) Many mistakes such as the following are found in your manuscript. Please recheck the whole manuscript carefully.

Line 66: Unnecessary space in front of period

Line 93: No period

Line 133: Sign between 50°C and 55°C do not make sense

Line 186 and 190: Inappropriate reference numbers

Line 193(Caption of Figure 1): Inappropriate reference number

Line 202: Disunity among "aging" and "ageing"

Equation (1): Three same expressions are lined up

Line 218: Expression of "the a "

Line 231: No period

Line 247 and 252: "Fig. 3-5" and "Fig. 3-7" are difficult to distinguish from "Figs. 3-5" and "Figs. 3-7"

Figure 3: The scales of upper figures are hidden

Figure 4: Not sure which figure corresponds to which condition

Line 268: "Figure 4" seems to be a mistake in "Figure 5"

Figures 6 and 7: Not sure which figure corresponds to which condition

Line 286(Caption of Figure 6): "100, 300, 600 and 1,000 of dry abrasive wear" seems to be a mistake in "100, 300, 600 and 1,000 cycles of dry abrasive wear"

Line 290(Caption of Figure 7): "100, 300, 600 and 1,000 of wet abrasive wear" seems to be a mistake in "100, 300, 600 and 1,000 cycles of wet abrasive wear"

Line 322(Caption of Figure 11): "dry" is unnecessary

Figure 12: The scales of upper figures are hidden

Figure 13: Not sure which figure corresponds to which condition

Line 370(Caption of Figure 15): "dry abrasion" seems to be a mistake in "dry and wet abrasion"

Equation (2): "smaler" seems to be a mistake in "smaller"

Table 4: Inconsistency with text (Contains non-significant results)

Line 428(Caption of Figure 18): No period with first sentence

Line 430: No chapter numer

Response:

Thank you for paying attention to the details. Consistency and correct writing are also important to us. We corrected all mistakes and changed the descriptions of the indicated Figures, simultaneously making appropriate corrections in the text.

Yours faithfully

Authors

Reviewer 4 Report

The article explores the Process of Abrasive Wear of Epoxy-Carbon Fabric Composites. As it stands, the article looks purely applied, but the potential interest for a wide range of readers is not obvious. Obviously, the results of applied work carried out by order of the Ministry of National Defense of Poland are presented. Unfortunately, the reviewer cannot recommend the article for publication, because:

  1. The authors initially give an extremely narrow, applied nature of the problem. Scientific research should offer some more global perspective.
  2. During the research, an extremely limited set of equipment was used (optical microscope and profilometer). That is, only the macro and meso surface parameters were investigated. Surface micro-parameters were not considered, chemical and phase analysis was not carried out.
  3. The article is poorly prepared, full of errors and misprints. The English language needs correction.

Some specific notes are presented below:

The abstract is too long (traditionally no more than 200 words). Typically, the literary style involves the use of impersonal forms (not "authors explored" or "I investigated", but "was investigated" or "explored").

Line 25: "temperature difference:" - incomprehensible highlighting of the text.

"temperature difference: 116.5 C" - degrees Celsius? - it is better to indicate "from ... to ..."

Line 26 m2 – superscript

English should be additionally adjusted in terms of style (I recommend a professional language service / proofreading). For example, line 61 "described relate to issues related to ..." or line 64 "These modifiers include such modifiers ..." or line 81 “composition of the composite “ is a teutology.

Line 74, 90,110 - incomprehensible highlighting

I did not find a reference to Table 1 in the text.

Line 110-115. "The literature ... suggests the use of the scanning electron microscopy (SEM) too" is an obvious fact. SEM is used to study the surface. "It is a very useful imaging technique that utilized a beam of electrons to acquire high magnification images of specimens" - is this written for teenagers or schoolchildren? Why is this information in a scientific article?

But the authors don't use SEM in their research! Why then write about it?

Line 116. "occurring on the territory of Poland" - and, for example, in the Czech Republic or Lithuania - other atmospheric conditions? Then these Poland-specific conditions must be justified and described. The authors write about "moderate climate of Poland" (which is difficult to argue with), but the conditions of the experiment - "operating temperature of –60 to +72 oC" - in my opinion, are not quite "moderate" climate. I doubt that there are aircraft produced specifically for "Poland's conditions" or, for example, "Germany's conditions." The flight temperature is usually around minus 50 degrees. Almost anywhere in the world. It is unlikely that the plane is designed exclusively for the Warsaw - Cracow flight. That is, for example, this plane will no longer be able to fly from Warsaw to Budapest? It seems to me that such a problem statement is somewhat strange. Typically, special aircraft operating conditions assume extreme climates (eg the Arctic or the Sahara Desert), but not a "moderate" climate. In my opinion, it would be correct to set the boundary conditions of the experiment (temperature: from ... to ..., humidity, etc.) and, for example, the duration of the simulated flight (for example, "short distance, 1-2 hours"). Moreover, the data in Table 3 are applicable for hundreds of flights in Russia, USA, Canada, Argentina, Chile and Europe.

Lines 186 and 190 and 193. What are references [455] and [456] [458]?

Figure 1. If the equipment is standard, it makes no sense to give its image (you can find it on the Internet without any problems). It is more useful to give a diagram of the experiment.

What's the point in FIG. 2. What can the reader understand by looking at this figure?

FIG. 3 Much more useful. And it is quite enough.

Lines 247 and 252. And where FIG. 3-5 and 3-7?

Standard designation of Figures: 3a 3b ...

Line 387. The strange title of the section is "3.3. Others analysis". What kind of analysis?

Author Response

Dear Reviewer,

Thank you for your review. We focused on analyzing this review and made the appropriate changes. We are grateful for help in shaping the new form of our publication. Below are our explanations. All changes in the text are marked in red colour.

Comment:

The article explores the Process of Abrasive Wear of Epoxy-Carbon Fabric Composites. As it stands, the article looks purely applied, but the potential interest for a wide range of readers is not obvious. Obviously, the results of applied work carried out by order of the Ministry of National Defense of Poland are presented. Unfortunately, the reviewer cannot recommend the article for publication, because:

The authors initially give an extremely narrow, applied nature of the problem. Scientific research should offer some more global perspective.

During the research, an extremely limited set of equipment was used (optical microscope and profilometer). That is, only the macro and meso surface parameters were investigated. Surface micro-parameters were not considered, chemical and phase analysis was not carried out.

Response:

We are continuing our research and want to publish more analyzes soon. Thank you very much for your constructive comments.

Comment:

The article is poorly prepared, full of errors and misprints.

Response:

It is important for us that the publication is consistent and written correctly. All comments indicated below have been corrected.

Comment:

The English language needs correction.

Response:

Immediately after substantive acceptance, we will forward the article for language correction via the editorial staff of Materials. The article will receive a certificate.

Comment:

Some specific notes are presented below:

The abstract is too long (traditionally no more than 200 words). Typically, the literary style involves the use of impersonal forms (not "authors explored" or "I investigated", but "was investigated" or "explored").

Response:

Abstract has been shortened.

Comment:

Line 25: "temperature difference:" - incomprehensible highlighting of the text.

Response:

All text highlights and color changes in text are the corrections made after the comments of the previous reviewer. They were required by the editors to stay visible. The introduced changes were approved by this reviewer.

Comment:

"temperature difference: 116.5 C" - degrees Celsius? - it is better to indicate "from ... to ..."

Response:

We have changed the record.

Comment:

Line 26 m2 – superscript

Response:

We corrected this mistake.

Comment:

English should be additionally adjusted in terms of style (I recommend a professional language service / proofreading). For example, line 61 "described relate to issues related to ..." or line 64 "These modifiers include such modifiers ..." or line 81 “composition of the composite “ is a teutology.

Response:

Thank you. We will use a professional language service.

Comment:

Line 74, 90,110 - incomprehensible highlighting

Response:

same as above: editorial recommendations.

Comment:

I did not find a reference to Table 1 in the text.

Response:

We have added a reference to Table 1 in line 55.

Comment:

Line 110-115. "The literature ... suggests the use of the scanning electron microscopy (SEM) too" is an obvious fact. SEM is used to study the surface. "It is a very useful imaging technique that utilized a beam of electrons to acquire high magnification images of specimens" - is this written for teenagers or schoolchildren? Why is this information in a scientific article?

Response:

We have deleted this sentence.

Comment:

But the authors don't use SEM in their research! Why then write about it?

Response:

In the introduction, we wrote about SEM due to one of the suggestions of a previous reviewer. But now we have shortened this part of the introduction

We have conducted surface tests using an optical microscope with digital image recording. In the article, we have included illustrative images. Unfortunately, we did not perform surface tests using an electron microscope that allows creating 3D images. We are sorry, but at this stage it will be difficult to reproduce samples subjected to all aspects of environmental and consumption. Unfortunately, at this time we have no possibility to make 3D images. Thank you for your attention, it is justified. We will use it in future publications and enrich them for qualitative analysis based on 3D images.

Comment:

Line 116. "occurring on the territory of Poland" - and, for example, in the Czech Republic or Lithuania - other atmospheric conditions? Then these Poland-specific conditions must be justified and described. The authors write about "moderate climate of Poland" (which is difficult to argue with), but the conditions of the experiment - "operating temperature of –60 to +72 oC" - in my opinion, are not quite "moderate" climate. I doubt that there are aircraft produced specifically for "Poland's conditions" or, for example, "Germany's conditions." The flight temperature is usually around minus 50 degrees. Almost anywhere in the world. It is unlikely that the plane is designed exclusively for the Warsaw - Cracow flight. That is, for example, this plane will no longer be able to fly from Warsaw to Budapest? It seems to me that such a problem statement is somewhat strange. Typically, special aircraft operating conditions assume extreme climates (eg the Arctic or the Sahara Desert), but not a "moderate" climate. In my opinion, it would be correct to set the boundary conditions of the experiment (temperature: from ... to ..., humidity, etc.) and, for example, the duration of the simulated flight (for example, "short distance, 1-2 hours"). Moreover, the data in Table 3 are applicable for hundreds of flights in Russia, USA, Canada, Argentina, Chile and Europe.

Response:

You have right. Thank you for this suggestion. We have deleted Polish aspects form the article. Please see the changes in chapters 2.1, 2.2, 2.3 and in Table 3.

Comment:

Lines 186 and 190 and 193. What are references [455] and [456] [458]?

Response:

We corrected these mistakes.

Comment:

Figure 1. If the equipment is standard, it makes no sense to give its image (you can find it on the Internet without any problems). It is more useful to give a diagram of the experiment.

Response:

Thank you. We have rebuild this Figure.

Comment:

What's the point in FIG. 2. What can the reader understand by looking at this figure?

Response:

We wondered over the removal of this Figure. However, it seems to us that the visual representation of the colour change of the sample surface gives a general view of the changes under the influence of UV-A, as described in the first paragraph of chapter 3.1.

If it would be better for the article, we give up this Figure, and then we will modify the description of the first paragraph.

Comment:

FIG. 3 Much more useful. And it is quite enough.

Response:

Thank you.

Comment:

Lines 247 and 252. And where FIG. 3-5 and 3-7?

Response:

Thank you for your suggestion. We corrected these mistakes.

Comment:

Standard designation of Figures: 3a 3b ...

Response:

We have changed the descriptions of the Figures.

Comment:

Line 387. The strange title of the section is "3.3. Others analysis". What kind of analysis?

Response:

The word „results” will be more adequate.

Yours faithfully

Authors

Round 2

Reviewer 3 Report

Thank you for your prompt response. However, your manuscript still seems inadequate. Please consider my comments (1) to (3) once again.

(1) Although your experimental results are interesting and characteristic, there is a problem with the completeness of the paper because there is no systematic and logical discussion on those results.
Detailed analysis may be left to another paper, but this paper should also have a basic discussion from the viewpoint of materials science and tribological analysis.

(2) If Rmax is used in your manuscript, the definition of Rmax and the difference from Rz in the current ISO should be explained to avoid confusion for readers.
It is strange that only Ra is explained in detail.

(3) In your manuscript, the description for Fig. 8 is "The changes occurred both in the process of dry friction where the intensity dropped (the coefficient of the variable decreased its value) and of wet friction, where there was the slight intensification of abrasive wear in successive cycles.", while the description for Fig. 9 is "On this basis it can be concluded that the impact of UV-A for a period of up to six months has a positive influence on composite resistance to abrasive wear during the process of friction in the water environment, and adversely during dry friction.". However, these explanations appear to be inconsistent, which could cause reader confusion.
For example, the following flow of the explanation may be easier for the reader to understand.
---
Under the wet condition in Fig. 8, the slope of the regression line of the data increases as the number of months of ageing changes from 0 to 2 to 4, but the distribution of the data converges on the low weight loss side.
Therefore, regarding the effect of UV exposure, although rapid progress of local abrasive wear may be induced by accelerated deterioration of the material, it is considered that change of material properties improves overall wear durability.
Under the wet condition of Fig. 9, since the weight loss decreases with the number of months of ageing, and it is considered that overall wear durability is improved as in the case of Fig. 8.
---
After reviewing the explanations of Fig. 8 and Fig. 9, and Fig. 14 and Fig. 15, and then please consider systematically and logically the difference in the tendency of weight loss between UV exposure and thermal shock.

Author Response

Dear Reviewer,

Thank you for your time to help us improve our article. 
We read the received review. We tried to refer to your helpful tips.

Comment:

Although your experimental results are interesting and characteristic, there is a problem with the completeness of the paper because there is no systematic and logical discussion on those results.

Detailed analysis may be left to another paper, but this paper should also have a basic discussion from the viewpoint of materials science and tribological analysis.

 Response:

Thank you for noticing that the research results are interesting. We discussed the results in our team once again. We tried to improve this aspect.

Comment:

If Rmax is used in your manuscript, the definition of Rmax and the difference from Rz in the current ISO should be explained to avoid confusion for readers.

It is strange that only Ra is explained in detail.

 Response:

Thank you for your comment. We have completed the missing definitions.

Comment:

In your manuscript, the description for Fig. 8 is "The changes occurred both in the process of dry friction where the intensity dropped (the coefficient of the variable decreased its value) and of wet friction, where there was the slight intensification of abrasive wear in successive cycles.", while the description for Fig. 9 is "On this basis it can be concluded that the impact of UV-A for a period of up to six months has a positive influence on composite resistance to abrasive wear during the process of friction in the water environment, and adversely during dry friction.". However, these explanations appear to be inconsistent, which could cause reader confusion.

For example, the following flow of the explanation may be easier for the reader to understand.

---

Under the wet condition in Fig. 8, the slope of the regression line of the data increases as the number of months of ageing changes from 0 to 2 to 4, but the distribution of the data converges on the low weight loss side.

Therefore, regarding the effect of UV exposure, although rapid progress of local abrasive wear may be induced by accelerated deterioration of the material, it is considered that change of material properties improves overall wear durability.

Under the wet condition of Fig. 9, since the weight loss decreases with the number of months of ageing, and it is considered that overall wear durability is improved as in the case of Fig. 8.

---

After reviewing the explanations of Fig. 8 and Fig. 9, and Fig. 14 and Fig. 15, and then please consider systematically and logically the difference in the tendency of weight loss between UV exposure and thermal shock.

 Response:

Thank you for the hint. We have attached your proposal to the article. We also slightly modified the description to Figures 14 and 15. In the description of Figures 14 and 15, we added another explanation of the differences in the tendency of weight loss between UV exposure and thermal shocks.

Thank you very much for your advice and help. We want this article to be the starting point for the next steps. We will soon write a new publication that will present the studied phenomena from a different perspective.

Kind regards

Authors

Reviewer 4 Report

Since the authors made a number of significant changes to the manuscript, it became more acceptable for publication. Given that the use of SEM allows for a deeper and more comprehensive study of processes. The reviewer insists on additional language service. After re-examination, the reviewer believes that this article is not complete and comprehensive, it can be improved and supplemented, but, at the same time, it is of interest to the reader in its current form, therefore it can be recommended for publication, provided that the English is improved.

Line 70 - typo - reference 276

Author Response

Dear Reviewer,

Thank you for your time to help us improve our article. We promise after receiving positive recommendations from all reviewers, we will send this article for linguistic proofreading via the editorial staff of Materials MDPI. The article will receive a certificate.

We corrected the reference number.

Please accept our gratitude for your cooperation.

Kind regards

Authors